# Beyond Diffusion: Consistency Models for One-Step, High-Fidelity MRI Reconstruction

**Mary-Brenda Akoda**[1]                                          M.AKODA24@ALUMNI.IMPERIAL.AC.UK
**Chen Qin**[2]                                                        C.QIN15@IMPERIAL.AC.UK

[1] *Department of Computing & I-X, Imperial College London, UK*

[2] *Department of Electrical and Electronic Engineering & I-X, Imperial College London, UK*

**Editors:** Accepted for publication at MIDL 2026

## Abstract

Magnetic resonance imaging (MRI) provides excellent soft-tissue contrast, but suffers from long acquisition times, limiting throughput and increasing patient discomfort. Diffusion-based generative models have recently achieved state-of-the-art reconstruction quality for accelerated MRI, but typically require hundreds to thousands of neural function evaluations (NFEs), which severely limits their practicality in time-sensitive clinical settings. We introduce C-MORE (Consistency-Model-based One-step REconstruction for MRI), to our knowledge, the first *one-step* consistency model framework for accelerated MRI reconstruction. C-MORE investigates an unconditional one-step prior and solves the inverse problem in one NFE by leveraging measurement-guided encoding and tunable physics-based refinement, thus eliminating multi-NFE diffusion sampling, while retaining a controllable quality-speed trade-off. On the MICCAI CMR×Recon dataset spanning multiple cardiac contrasts and both single- and multi-coil acquisitions, C-MORE outperforms state-of-the-art diffusion-based samplers and strong non-diffusion unrolled methods across accelerations in just 1 NFE, while reconstructing images in $0.18 - 0.52$ s ($\approx 22 - 193\times$ faster than diffusion-based methods requiring hundreds of NFEs). Remarkably, without any retraining or finetuning, C-MORE also demonstrates cross-anatomy generalisation to the *unseen* fastMRI knee dataset from NYU Langone Health and Facebook AI Research, again surpassing state-of-the-art methods across accelerations. These results establish C-MORE as a practical blueprint for real-time, high-fidelity MRI reconstruction across diverse contrasts, acquisition settings, anatomies, and accelerations.

**Keywords:** MRI reconstruction, diffusion models, consistency models, inverse problems, cardiac MRI.

## 1. Introduction

Magnetic resonance imaging (MRI) provides excellent soft-tissue contrast without ionising radiation, but conventional scans remain slow, often between fifteen minutes and over an hour and a half, leading to patient discomfort, motion artefacts, and reduced clinical throughput. Accelerated MRI reduces scan time by undersampling $k$-space (the frequency domain), but reconstructing an image from sparse measurements is an ill-posed inverse problem that requires strong priors. This challenge is particularly pronounced in quantitative parametric mapping, where each slice is acquired at multiple inversion or echo times to estimate T1/T2 maps: a single examination yields a sequence of contrast images per slice,

and training separate reconstruction models for each contrast is inefficient and difficult to scale in practice.

Classical approaches such as parallel imaging (e.g. SENSE, GRAPPA) and compressed sensing (CS) exploit coil sensitivities or sparsity-promoting regularisers to stabilise the inverse problem (Lustig et al., 2007). While effective, these methods rely on hand-crafted priors, require lengthy optimisations, and can be sensitive to noise, parameter tuning, and the choice of sampling pattern. Deep learning-based reconstructions, including unrolled variational networks and image-domain convolutional neural networks (CNNs), learn powerful data-driven priors and can produce high-quality reconstructions in a single forward pass (Sriram et al., 2020; Hammernik et al., 2018). However, such models are typically trained for specific masks, acceleration factors, and acquisition protocols; their performance can degrade when the sampling pattern or contrast distribution shifts, often necessitating retraining or maintaining multiple models for different protocols.

Generative models have recently emerged as strong priors for accelerated MRI. Generative adversarial network (GAN)-based methods encourage realistic image appearance but can be unstable and hallucinate anatomically incorrect details (Mardani et al., 2019; Yang et al., 2018). Diffusion models (DMs) provide a more principled generative formulation, learning a time-dependent score or denoiser that can be combined with the MRI acquisition operator to perform posterior sampling. In score-based diffusion models for accelerated MRI, Chung and Ye (Chung and Ye, 2022) train a continuous-time score network on fully sampled images and alternate between stochastic or deterministic diffusion updates and $k$-space data consistency (DC) projections during sampling. Related work extends diffusion posterior sampling and plug-and-play guidance to MRI and other inverse problems, typically interleaving denoising steps with explicit DC updates at each iteration (Chung et al., 2023; Peng et al., 2022). More recently, Malyala et al. proposed SPA-MRI, which adapts consistency weights within a diffusion sampler to improve robustness across sampling patterns and contrasts in cardiac MRI (Malyala et al., 2024). Despite their strong reconstruction quality, these diffusion-based approaches generally require hundreds to thousands of neural function evaluations (NFEs) per image, with one or more DC steps at each iteration, which limits their practicality for time-sensitive clinical workflows.

Consistency models (CMs) have been proposed as a way to achieve quality comparable to diffusion models while enabling one- or few-step sampling (Song et al., 2023). Instead of integrating many small denoising updates along a diffusion trajectory, a CM learns a direct mapping from a noisy state to a clean state that is consistent with the probability-flow ordinary differential equation (ODE); hence, samples can be generated in a single or small number of NFEs. When trained either by distillation from a pre-trained diffusion model (consistency distillation) or directly from data (consistency training), CMs can approach the performance of diffusion models on natural image benchmarks with one to a few NFEs, offering a promising route towards real-time inverse problems. However, while CM applications to inverse problems are beginning to emerge (Zhao et al., 2024; Garber and Tirer, 2025), prior approaches still rely on few-step sampling or generic inverse problem setups.

In parallel, guided generation methods augment diffusion or consistency model backbones with auxiliary encoders that inject conditioning signals. ControlNet, for example, attaches a parallel encoder with zero-initialised lateral connections to steer the generative process using edge maps, depth maps, or other hints, without retraining the backbone

(Zhang et al., 2023). Building on these ideas, Zhao et al. combine a CM prior with a ControlNet-style encoder and an explicit projection step to address generic inverse problems with one or few NFEs (Zhao et al., 2024). Their formulation, however, is developed for general imaging tasks: it does not exploit MRI-specific acquisition operators, nor is it tailored to multi-contrast parametric mapping scenarios where a single model must handle multiple contrasts per slice under stringent runtime constraints. Moreover, to refine results, their reconstruction pipeline still relies on multistep sampling interleaved with DC updates, and reports no further improvement beyond two NFEs, thereby limiting performance.

In this work, we investigate a one-step CM tailored to accelerated MRI to achieve high-fidelity reconstructions in one NFE while generalising across contrasts, acquisition settings, and acceleration factors. Specifically, we propose C-MORE (Consistency-Model-based One-step Reconstruction for MRI), a one-step generative reconstruction framework that builds on an unconditional CM prior and leverages an MRI-specific reconstruction strategy based on measurement-guided encoding and tunable conjugate gradient (CG) data consistency. Rather than alternating denoising and DC steps as in prior literature, C-MORE concentrates all learned generative reasoning into one guided CM evaluation and then performs multiple purely physics-based refinement updates. This design collapses the hundreds to thousands of DC corrections used in diffusion samplers into a single tunable $K$-CG block (where $K$ is the maximum number of refinement updates); this tunable $K$ enables a test-time trade-off between runtime and fidelity while still retaining one NFE and sub-second runtime.

Our contributions can be summarised as follows. 1) We propose, to our knowledge, the first *one-step* consistency model framework for accelerated MRI reconstruction. 2) We introduce a hybrid reconstruction strategy that leverages measurement-guided encoding with tunable $K$-CG refinement updates, eliminating the need for multi-NFE diffusion sampling. 3) In just one NFE, a single C-MORE model reconstructs across contrasts, acquisition settings, acceleration factors, and anatomies without retraining, achieving higher reconstruction quality than state-of-the-art diffusion-based samplers and strong non-diffusion unrolled methods, while running in $0.18 - 0.52\,\mathrm{s}$ on average ($\approx 22 - 193\times$ speed-up compared to state-of-the-art diffusion-based methods requiring hundreds of NFEs).

## 2. Methods

### 2.1. Problem Formulation and Framework Overview

Accelerated MRI reconstruction can be formalised as recovering an image $x \in \mathbb{C}^N$ from noisy, undersampled $k$-space measurements $y$, thus:

$$y \;=\; Ax + \eta, \tag{1}$$

where $A : \mathbb{C}^N \to \mathbb{C}^M$ is the MRI acquisition operator and $\eta$ models measurement noise; $A^{\mathrm{H}}$ denotes its adjoint. For a single-coil acquisition, $A$ typically combines a sampling mask $M$ and a discrete Fourier transform $F$, giving $y = MFx + \eta$. For multi-coil MRI, each coil $c$ acquires $y_c = MFS_c x + \eta_c$ where $S_c$ encodes coil sensitivities; stacking all coils yields $y$ and the associated operator $A$.

Our goal is to recover a high-fidelity MR image $x$ from the undersampled acquired k-space data $y$ (Eq. (1)). However, this inverse problem is ill-posed ($M \ll N$), requiring

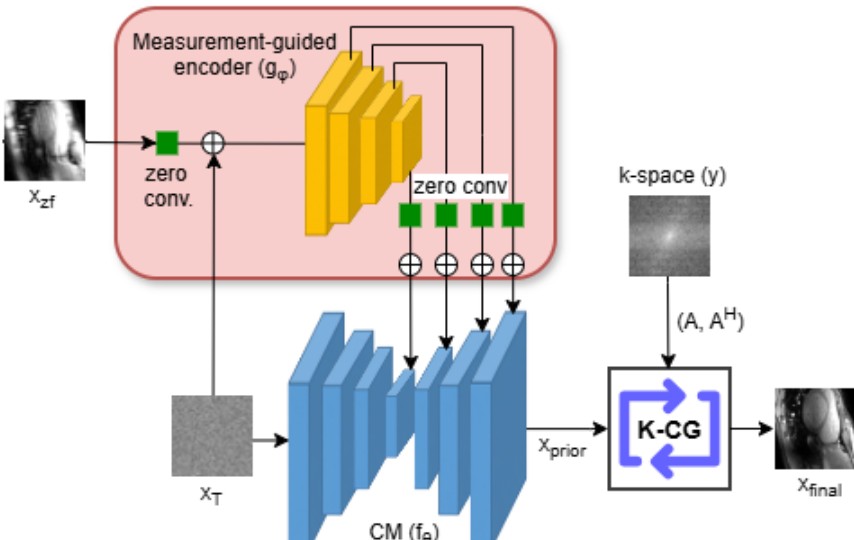

Figure 1: Overview of the proposed C-MORE framework. A pre-trained EDM teacher is distilled into a one-step CM prior $f_\theta$. During reconstruction, a measurement-guided encoder $g_\varphi$ conditions the distilled one-step CM $f_\theta$ on $x_{\mathrm{zf}}$ via zero-initialised lateral connections (green), and a subsequent CG block performs $K$ physics-based refinement updates using the MRI forward operator, enforcing $k$-space consistency while keeping NFEs fixed at one.

strong priors. Fig. 1 provides an overview of our proposed C-MORE framework. C-MORE comprises four main components: (i) an unconditional high-fidelity diffusion teacher trained to model the distribution of fully sampled MR images; (ii) an unconditional one-step consistency model (CM) prior $f_\theta$ obtained by distilling the teacher; (iii) a measurement-guided encoder that processes the zero-filled reconstruction $x_{\mathrm{zf}}$, obtained by applying the inverse Fourier transform, and injects its features into the CM, allowing us to exploit measurement-specific information at inference time without retraining the prior for a specific $A$; and (iv) a conjugate gradient (CG) refinement module that performs $K$ physics-based updates, thus enforcing $k$-space consistency with tunable $K$ strength.

## 2.2. Unconditional Diffusion Teacher Training

To construct a high-fidelity one-step prior, we first train an unconditional Elucidated Diffusion Model (EDM) (Karras et al., 2022) on fully sampled complex-valued MR images. EDM is a continuous-time diffusion formulation that parameterises the corruption process by a noise scale $\sigma \in [\sigma_{\min}, \sigma_{\max}]$ rather than a discrete time index. Compared to earlier discrete-time diffusion models, EDM introduces (i) a tailored noise-level distribution, (ii) a preconditioning scheme that stabilises training and sampling across $\sigma$, and (iii) principled choices of parameterisation, loss weighting, and solver discretisation. This design thus yields state-of-the-art image quality with relatively few sampling steps (Karras et al., 2022), making EDM a strong candidate to serve as a teacher in our consistency distillation pipeline.

Let $x_0 \sim p_{\mathrm{data}}$ denote a fully sampled MR image and $z \sim \mathcal{N}(0, I)$, standard Gaussian noise. EDM perturbs $x_0$ by sampling a noise level $\sigma$ from a fixed distribution $p(\sigma)$ and

forming: $x_\sigma = x_0 + \sigma z$. A U-Net denoiser $g_\phi$ takes $(x_\sigma, \sigma)$ as input and predicts a denoised sample. Following Karras et al. (2022), we train $g_\phi$ with a weighted denoising objective:

$$\mathcal{L}_{\mathrm{EDM}}(\phi) = \mathbb{E}_{x_0, \sigma, z}\big[\lambda(\sigma)\,\|g_\phi(x_\sigma, \sigma) - x_0\|_2^2\big], \tag{2}$$

where $\lambda(\sigma)$ compensates for the varying difficulty of denoising at different noise levels. We represent complex-valued MR images as two channels (real and imaginary) and train $g_\phi$ unconditionally on fully sampled MR images—not dependent on sampling mask, acceleration factor, or coil configuration.

The EDM thus provides a high-quality unconditional generative prior over fully sampled MR images. In the subsection that follows, we treat $g_\phi$ as a fixed teacher and use its probability-flow ordinary differential equation (ODE) trajectory as the target for consistency distillation. Reconstruction pipelines based on $g_\phi$ combined with various data consistency schemes serve as strong diffusion baselines in our experiments (Section 3); here, however, we focus solely on its role as a teacher.

## 2.3. Perceptual Consistency Distillation

We obtain an unconditional one-step CM prior $f_\theta$ via consistency distillation of our high-fidelity EDM teacher $g_\phi$ into $f_\theta$. Let $\{t_n\}_{n=0}^N$ denote a discretisation of the continuous time interval $[0, 1]$, with $t_0 = 0$ (clean image) and $t_N = 1$ (noisy sample). The EDM teacher defines a reverse trajectory under its probability-flow ODE, which we approximate using a Heun integrator $\phi$. For neighbouring times $t_{n+1} > t_n$, the teacher state at $t_{n+1}$ is denoted $x_{t_{n+1}}$, and a single Heun update yields a one-step prediction $\hat{x}_{t_n}^\phi$ of the earlier state $x_{t_n}$. Consistency distillation then trains $f_\theta$ such that, given $x_{t_{n+1}}$, it outputs a state aligned with $\hat{x}_{t_n}^\phi$. Following the online-target scheme of Song et al. (2023), we maintain an exponential moving average (EMA) copy of the parameters $\theta^-$ and optimise:

$$\mathcal{L}_{\mathrm{CD}}(\theta, \theta^-; \phi) = \mathbb{E}\big[\lambda(t_n)\,d\big(f_\theta(x_{t_{n+1}}, t_{n+1}), f_{\theta^-}(\hat{x}_{t_n}^\phi, t_n)\big)\big], \tag{3}$$

where $\lambda(t_n)$ is a time-dependent weight and $d(\cdot, \cdot)$ is a distance metric between the online CM output and its EMA counterpart. This objective collapses the multi-step EDM reverse trajectory into a single, globally consistent mapping. For MR images, a purely pixel-wise distance can lead to overly smooth one-step samples since a single U-Net must capture the entire denoising trajectory. Thus, to better preserve local structure, we adopt a perceptual distance based on the Learned Perceptual Image Patch Similarity (LPIPS) metric (Zhang et al., 2018), applied to the magnitude images of the complex-valued outputs. Substituting this choice of $d$ into Eq. (3) yields a perceptual consistency distillation objective that aligns the CM with the EMA target in a feature space that is sensitive to structural similarity. This choice of $d$ is also consistent with prior CM literature, where LPIPS was shown to substantially outperform $\ell_1$ and $\ell_2$ losses for image-domain consistency distillation (Song et al., 2023). In practice, we find that this design helps maintain anatomical structures in one-step reconstructions.

After training, the CM defines an unconditional one-step prior: given a single noise sample $x_T$ at the largest noise level, we obtain an MR image in one forward pass: $x_0 = f_\theta(x_T, T)$. This design allows the same CM to be reused across settings, since acquisition-specific information is injected only at inference time (Section 2.4).

## 2.4. One-Step Guided Inference with Tunable Data Consistency

The CM prior described in Section 2.3 captures the distribution of fully sampled MR images but, by construction, is independent of the specific measurement $y$. Consequently, state-of-the-art diffusion-based methods typically solve the inverse problem via posterior sampling, which requires a high number of NFEs and leads to long reconstruction times. To address this limitation, we propose a hybrid reconstruction strategy that: (i) introduces measurement awareness into the generative step via measurement-guided encoding, avoiding the need to retrain the CM or compromise its unconditional, contrast-agnostic nature, and (ii) incorporates tunable physics-based refinement updates, eliminating the need for multi-NFE diffusion sampling.

### 2.4.1. Measurement-guided encoding

Let $x_{\mathrm{zf}}$ denote the zero-filled reconstruction, obtained by applying the inverse Fourier transform and representing the result as a two-channel complex image (real and imaginary parts). Although $x_{\mathrm{zf}}$ contains aliasing artefacts, it already encodes coarse anatomy, contrast information, and coil sensitivity effects. Our goal is to use $x_{\mathrm{zf}}$ to steer the one-step CM towards solutions that are both likely under the prior and compatible with the measured data, without altering the unconditional CM training.

To this end, we attach a lightweight measurement-guided encoder to the CM in the spirit of ControlNet-style conditioning (Zhang et al., 2023; Zhao et al., 2024). A parallel encoder $g_\varphi$ with the same multi-scale structure as the CM backbone processes $x_{\mathrm{zf}}$ and produces feature maps $\{h_\ell\}$ at multiple resolutions. At each encoder block $\ell$ of the CM, we fuse these features into the CM activations via a zero-initialised 1×1 convolution:

$$u_\ell = \mathrm{Enc}_\ell^{\mathrm{CM}}(x_T, t), \quad h_\ell = \mathrm{Enc}_\ell^{\mathrm{enc}}(x_{\mathrm{zf}}), \quad u'_\ell = u_\ell + W_\ell h_\ell, \qquad W_\ell\big|_{\mathrm{init}} = 0, \qquad (4)$$

where $u_\ell$ are the original CM features, $u'_\ell$ the modified features passed to subsequent CM layers and the decoder, and $W_\ell$ are the learnable 1×1 fusion convolutions.

During training of this measurement-guided encoder, we freeze the CM parameters $\theta$ and optimise only $\varphi$ and $\{W_\ell\}$ using paired fully sampled / undersampled data. Training minimises a reconstruction loss between the encoder-guided CM output and the fully sampled reference image, encouraging the encoder to use $x_{\mathrm{zf}}$ to refine the one-step sample towards the correct anatomy and contrast while leaving the unconditional CM backbone unchanged. Because the fusion weights $W_\ell$ are initialised to zero, the network initially reproduces the unconditional CM output and gradually learns to incorporate measurement information. The resulting measurement-aware one-step sample is:

$$x_{\mathrm{prior}} = f_\theta(x_T, T \mid x_{\mathrm{zf}}), \qquad (5)$$

where the conditioning enters only through the additional encoder and lateral connections.

### 2.4.2. $K$ Physics-based refinement

The guided CM output $x_{\mathrm{prior}}$ incorporates information from $x_{\mathrm{zf}}$ but is not guaranteed to satisfy the acquisition operator in Eq. (1). To improve data fidelity, we add a final conjugate gradient (CG) refinement stage that solves for the image $x$ in the least-squares problem:

$$A^{\mathrm{H}} A x = A^{\mathrm{H}} y \qquad (6)$$

using $x_{\text{prior}}$ as the initial iterate and a fixed maximum number of updates, $K$. Each update requires only applications of $A$ and $A^{\text{H}}$, implemented efficiently via Fast Fourier transforms (FFTs) and sensitivity maps (for multi-coil). We stop after $K$ updates or earlier if the residual norm $\|Ax - y\|_2$ falls below a preset tolerance, and denote the final iterate by $x_{\text{final}}$.

The maximum update count $K$ acts as a test-time knob that trades runtime against the strength of data-fidelity enforcement without retraining: small $K$ values preserve the CM output but apply only mild corrections, whereas larger $K$ values drive the solution closer to the least-squares solution of Eq. (6) at the cost of additional matrix-vector operations. In contrast to posterior-sampling approaches that interleave one or a few data consistency updates with every network evaluation (Chung and Ye, 2022; Chung et al., 2023; Peng et al., 2022; Chung et al., 2024; Wang et al., 2023; Malyala et al., 2024; Zhao et al., 2024), C-MORE concentrates most of its data-fidelity enforcement into a single CG refinement stage after one measurement-guided CM evaluation. This strategy is more efficient considering diffusion-based posterior-sampling methods accumulate hundreds to thousands of DC steps across many NFEs, whereas our design uses a cumulatively smaller number of maximum updates ($K \in [100, 200]$ in our experiments), without increasing the number of NFEs and while keeping the refinement purely physics-based and independent of the generative prior.

### 2.5. C-MORE (Consistency-Model-based One-step REconstruction)

Combining the components above, C-MORE performs one-step reconstruction as in Algorithm 1: Given a measurement $y$, we first form the zero-filled reconstruction, then sample a single noise realisation $x_T$ at the largest EDM noise level and run the measurement-guided CM to obtain the prior estimate $x_{\text{prior}}$. Finally, we run $K$ updates of CG on Eq. (6), initialised at $x_{\text{prior}}$, to obtain the refined reconstruction $x_{\text{final}}$. In this way, all learned generative reasoning is concentrated into a single guided CM evaluation, and subsequent updates are governed solely by the MRI acquisition operator, thus achieving one-to-two orders of magnitude speed-ups over diffusion samplers that require 100–1000 NFEs.

---

**Algorithm 1:** C-MORE (Consistency-Model-based One-step REconstruction)

---

**Input:** Measurement $y$, sampling mask $M$, Fourier operator $F$, acquisition operator $A$ and its adjoint $A^{\text{H}}$, CM prior $f_\theta$, measurement encoder $g_\varphi$, maximum number of CG updates $K$, maximum noise level $\sigma_{\max}$

**Output:** $x_{\text{final}}$, the reconstructed MR image

$x_{\text{zf}} \leftarrow F^{-1}(M^\top y)$                                     `// Zero-filled reconstruction`

Sample $x_T \sim \mathcal{N}(0, \sigma_{\max}^2 I)$

$x_{\text{prior}} \leftarrow f_\theta(x_T, T \mid x_{\text{zf}})$                            `// Measurement-guided CM`

$x_{\text{final}} \leftarrow \text{CG}\big(A^{\text{H}}A, A^{\text{H}}y, x_{\text{prior}}, K\big)$

**return** $x_{final}$

---

## 3. Experiments and Results

**Dataset:** We evaluate on the MICCAI CMR×Recon 2023 dataset (Wang et al., 2024), which contains single- and multi-coil cardiac MR images for T1 and T2 mapping. Fully

sampled complex-valued images are retrospectively undersampled to simulate accelerated acquisitions. We use 15,576 complex 2D images (real and imaginary channels) for training and 7,758 undersampled images for testing, covering 9 T1-weighted and 3 T2-weighted contrasts per slice and padded to $160 \times 512$ to match model input. A single C-MORE model is trained across all T1/T2 mapping contrasts, and results are reported on a multi-coil test set. We consider acceleration factors $R \in \{4, 8, 10\}$ with uniform Cartesian undersampling masks provided with the dataset. These masks are identical across contrasts for a given acceleration factor. We also include a generalisability test that evaluates C-MORE on the fastMRI dataset (Knoll et al., 2020) (padded to $320 \times 512$ and using different masks across slices) to assess generalisability to unseen data.

**Baselines and Metrics:** To evaluate the effectiveness of C-MORE, we first compare against four state-of-the-art diffusion-based samplers, each applied using the same pre-trained EDM prior alongside their respective posterior sampling schemes: DDNM (Wang et al., 2023), SPA-MRI (Malyala et al., 2024), DPS (Chung et al., 2023), and two variants of DDS (Chung et al., 2024). For the first variant, we adapt DDS using EDM as its prior with CG at $K=5$ per NFE for 100 NFEs (500 total DC updates); we refer to this enhanced configuration as the *Teacher*. For the second variant, which we call the *Optimised Teacher*, we redistribute the same total number of DC updates by increasing the teacher's per-step budget to $K=20$ and reducing the trajectory to 25 NFEs, yielding an equivalent cumulative number of solver updates. All baselines are configured for high-quality reconstructions, with NFEs between 100 and 1000 for quantitative and qualitative experiments and DC enforcement at each NFE, reflecting typical settings in the literature. We evaluate C-MORE ("Ours") at $K=100$ and $K=200$. Peak signal-to-noise ratio (PSNR, dB), structural similarity index measure (SSIM), NFE, and runtime in seconds (measured on a single NVIDIA A5000 GPU) are used as evaluation metrics for comparisons.

In addition, to contextualise performance against fast non-diffusion learned reconstruction methods, we include MoDL (Aggarwal et al., 2018) as a representative unrolled baseline. Since MoDL is typically trained for a fixed acquisition setting, we train two separate MoDL models: one for multi-coil T1/T2 at $R=4$ and another for multi-coil T1/T2 at $R=8$. MoDL therefore requires a dedicated model per setting; we did not train a $10\times$ MoDL model in our experiments as it already showed inferior performance at $R=8$.

**Reconstruction Performance:** Table 1 summarises contrast-wise reconstruction quality and runtime for multi-coil settings across T1 and T2 at $R=4, 8, 10$. C-MORE with $K=200$ consistently achieves the highest PSNR across all contrasts and acceleration factors. Even in SSIM, C-MORE surpasses the Optimised Teacher—which requires 500 cumulative CG updates—in all cases except T1 at $4\times$ and $10\times$, where C-MORE remains a very close second (0.9925 vs. 0.9935 and 0.9722 vs. 0.9760) while still maintaining a single NFE.

In terms of efficiency, C-MORE runs in 0.18–0.52 s per image, compared to 11.48–34.72 s for diffusion baselines that require hundreds of NFEs, thus yielding speed-ups of $\approx$22–193$\times$. Even the Optimised Teacher is still slower ($\approx$2.3 s/image) than C-MORE's sub-second runtime. Crucially, C-MORE's gains are achieved without increasing NFE beyond 1. Increasing $K$ from 100 to 200 adds negligible cost (from 0.09–0.24 s to 0.16–0.39 s on average), and with residual-based early stopping at a tolerance of $10^{-8}$, some $K=200$ runs terminate in similar time to $K=100$.

Table 1: Contrast-wise PSNR, SSIM, NFEs, and per-contrast runtime for T1 and T2 at $4\times$, $8\times$, and $10\times$ undersampling (multi-coil test set). Values are reported as mean $\pm$ standard deviation across the test set. Best results are in **bold**; second-best are underlined.

| Acc | Method | NFE | T1 | | | T2 | | |
|---|---|---|---|---|---|---|---|---|
| | | | PSNR ↑ | SSIM ↑ | Runtime (s) ↓ | PSNR ↑ | SSIM ↑ | Runtime (s) ↓ |
| $4\times$ | **Non-diffusion (unrolled)** | | | | | | | |
| | MoDL | 1 | $44.78 \pm 3.46$ | $0.9859 \pm 0.0088$ | $\mathbf{0.09} \pm 0.05$ | $42.46 \pm 2.95$ | $0.9869 \pm 0.0060$ | $\mathbf{0.09} \pm 0.05$ |
| | **Diffusion-based samplers** | | | | | | | |
| | DPS | 500 | $44.86 \pm 2.87$ | $0.9827 \pm 0.0102$ | $23.53 \pm 0.22$ | $41.77 \pm 2.19$ | $0.9804 \pm 0.0126$ | $23.50 \pm 0.21$ |
| | SPA-MRI | 500 | $40.18 \pm 6.15$ | $0.9441 \pm 0.0514$ | $34.70 \pm 15.11$ | $39.09 \pm 3.59$ | $0.9629 \pm 0.0344$ | $22.22 \pm 0.24$ |
| | DDNM | 100 | $38.03 \pm 2.29$ | $0.9240 \pm 0.0279$ | $11.48 \pm 0.30$ | $36.47 \pm 1.65$ | $0.9274 \pm 0.0367$ | $11.48 \pm 0.27$ |
| | DDS (Teacher, $K{=}5$) | 100 | $48.44 \pm 0.97$ | $0.9920 \pm 0.0057$ | $17.14 \pm 2.40$ | $46.56 \pm 1.52$ | $\underline{0.9941} \pm 0.0047$ | $11.38 \pm 0.23$ |
| | Optimised Teacher ($K{=}20$) | 25 | $48.90 \pm 0.68$ | $\mathbf{0.9935} \pm 0.0055$ | $2.36 \pm 0.42$ | $\underline{48.90} \pm 0.68$ | $0.9935 \pm 0.0055$ | $2.36 \pm 0.42$ |
| | C-MORE ($K{=}100$) | 1 | $\underline{49.05} \pm 0.45$ | $0.9925 \pm 0.0051$ | $\underline{0.37} \pm 0.12$ | $48.81 \pm 0.57$ | $\mathbf{0.9965} \pm 0.0030$ | $\mathbf{0.18} \pm 0.05$ |
| | C-MORE ($K{=}200$) | 1 | $\mathbf{49.06} \pm 0.45$ | $\underline{0.9925} \pm 0.0051$ | $\mathbf{0.31} \pm 0.11$ | $\mathbf{48.92} \pm 0.48$ | $\mathbf{0.9965} \pm 0.0030$ | $\underline{0.24} \pm 0.05$ |
| $8\times$ | **Non-diffusion (unrolled)** | | | | | | | |
| | MoDL | 1 | $35.91 \pm 2.71$ | $0.9305 \pm 0.0243$ | $\mathbf{0.09} \pm 0.03$ | $35.81 \pm 1.89$ | $0.9438 \pm 0.0204$ | $\mathbf{0.09} \pm 0.03$ |
| | **Diffusion-based samplers** | | | | | | | |
| | DPS | 500 | $38.95 \pm 4.20$ | $0.9437 \pm 0.0313$ | $23.54 \pm 0.22$ | $36.59 \pm 2.48$ | $0.9490 \pm 0.0213$ | $23.52 \pm 0.20$ |
| | SPA-MRI | 500 | $34.59 \pm 4.58$ | $0.8840 \pm 0.0605$ | $34.71 \pm 15.12$ | $33.85 \pm 3.07$ | $0.9221 \pm 0.0367$ | $22.24 \pm 0.25$ |
| | DDNM | 100 | $36.14 \pm 2.61$ | $0.9050 \pm 0.0334$ | $11.49 \pm 0.29$ | $34.79 \pm 1.84$ | $0.9128 \pm 0.0397$ | $11.47 \pm 0.27$ |
| | DDS (Teacher, $K{=}5$) | 100 | $43.03 \pm 2.91$ | $0.9754 \pm 0.0100$ | $17.12 \pm 2.40$ | $40.15 \pm 2.04$ | $0.9750 \pm 0.0090$ | $11.38 \pm 0.24$ |
| | Optimised Teacher ($K{=}20$) | 25 | $\underline{44.61} \pm 2.36$ | $\underline{0.9846} \pm 0.0064$ | $2.33 \pm 0.41$ | $\underline{44.61} \pm 2.36$ | $\underline{0.9846} \pm 0.0064$ | $2.33 \pm 0.41$ |
| | C-MORE ($K{=}100$) | 1 | $43.85 \pm 2.31$ | $0.9761 \pm 0.0109$ | $\mathbf{0.42} \pm 0.13$ | $42.49 \pm 1.96$ | $0.9842 \pm 0.0080$ | $\mathbf{0.18} \pm 0.05$ |
| | C-MORE ($K{=}200$) | 1 | $\mathbf{47.20} \pm 1.59$ | $\mathbf{0.9885} \pm 0.0064$ | $\underline{0.52} \pm 0.15$ | $\mathbf{46.36} \pm 1.58$ | $\mathbf{0.9933} \pm 0.0045$ | $\underline{0.35} \pm 0.05$ |
| $10\times$ | **Diffusion-based samplers** | | | | | | | |
| | DPS | 500 | $36.22 \pm 4.22$ | $0.9243 \pm 0.0407$ | $23.52 \pm 0.22$ | $35.38 \pm 2.51$ | $0.9401 \pm 0.0231$ | $23.51 \pm 0.21$ |
| | SPA-MRI | 500 | $32.55 \pm 3.97$ | $0.8574 \pm 0.0635$ | $34.72 \pm 15.12$ | $32.98 \pm 3.05$ | $0.9167 \pm 0.0332$ | $22.23 \pm 0.26$ |
| | DDNM | 100 | $35.32 \pm 2.71$ | $0.8985 \pm 0.0348$ | $11.49 \pm 0.30$ | $34.44 \pm 1.90$ | $0.9131 \pm 0.0395$ | $11.47 \pm 0.29$ |
| | DDS (Teacher, $K{=}5$) | 100 | $41.21 \pm 3.09$ | $0.9678 \pm 0.0133$ | $17.12 \pm 2.40$ | $38.89 \pm 2.10$ | $0.9696 \pm 0.0108$ | $11.39 \pm 0.23$ |
| | Optimised Teacher ($K{=}20$) | 25 | $\underline{42.20} \pm 2.70$ | $\mathbf{0.9760} \pm 0.0087$ | $2.31 \pm 0.40$ | $\underline{42.20} \pm 2.70$ | $\underline{0.9760} \pm 0.0087$ | $2.31 \pm 0.40$ |
| | C-MORE ($K{=}100$) | 1 | $40.07 \pm 2.52$ | $0.9505 \pm 0.0166$ | $\mathbf{0.41} \pm 0.12$ | $39.29 \pm 1.88$ | $0.9700 \pm 0.0104$ | $\mathbf{0.18} \pm 0.05$ |
| | C-MORE ($K{=}200$) | 1 | $\mathbf{43.25} \pm 2.42$ | $\underline{0.9722} \pm 0.0124$ | $\underline{0.52} \pm 0.15$ | $\mathbf{42.54} \pm 1.85$ | $\mathbf{0.9838} \pm 0.0077$ | $\underline{0.36} \pm 0.04$ |

To further contextualise C-MORE's competitiveness against the teacher, it is worth noting the disparity in training budgets: the EDM teacher was trained for $\approx$750,000 training iterations, compared to the CM student's $\approx$200,000 ($\approx$27% of the teacher's training budget). Despite this smaller budget, C-MORE delivers one-NFE reconstruction at test time and surpasses the teacher in our experiments (particularly at $K{=}200$).

Finally, to compare against non-diffusion methods: MoDL, a representative fast unrolled baseline, achieves very low per-image runtimes ($\approx$0.09 s) but requires separate models for each acquisition setting. Furthermore, while MoDL is efficient at lower acceleration (e.g., at $4\times$, it attains SSIM/PSNR competitive with some diffusion baselines), its reconstruction quality degrades substantially at higher accelerations (e.g., T1 PSNR drops from $\approx$44.78 dB at $4\times$ to $\approx$35.91 dB at $8\times$; see Table 1). By contrast, a single C-MORE model trained across contrasts and coil settings maintains high fidelity across $R \in \{4, 8, 10\}$, better satisfying the combined objective of state-of-the-art reconstruction quality, protocol generalisability, and sub-second runtime.

Fig. 2 shows qualitative reconstructions at $R{=}4, 8, 10$. The zero-filled images exhibit severe aliasing at higher accelerations. Among the baselines, the teacher and a DDNM variant using CG for further enhancement yield visually high-quality reconstructions but require 100 NFEs and 500 cumulative CG updates ($K{=}5$/NFE), while DPS degrades with

a higher iteration count (1000 NFEs). In contrast, C-MORE ($K$=100 and $K$=200) shows no visible artefacts, while running in one NFE and sub-second time. Full T1 and T2 parametric-sequence reconstructions are provided in Appendix A.

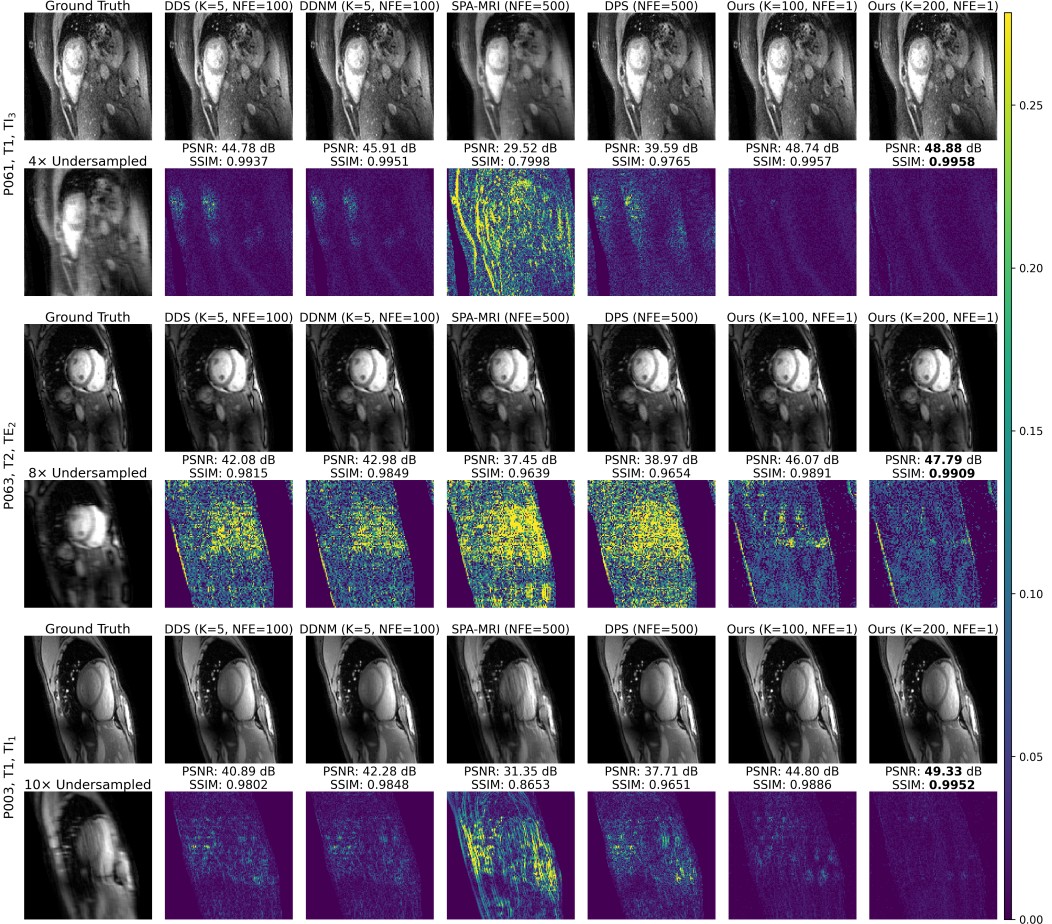

Figure 2: Cardiac MR reconstructions at $4\times$, $8\times$, $10\times$ across different patients and tissue contrasts, acquired at varying inversion times (TI) and echo times (TE). We compare ground truth, zero-filled (undersampled), teacher ("DDS") and a DDNM variant (both using CG at K=5/NFE for further enhancement), SPA-MRI, DPS, and C-MORE ("Ours") at $K \in [100, 200]$, plus error maps. Best results are in **bold** underneath the reconstructions.

**Ablation Studies:** To analyse the choice of refinement operator and the effect of varying $K$, we first fix the CM (without measurement-guided encoding) and vary $K$ for different posterior sampling schemes at $R$=10 while keeping NFE= 1. For $K$ from 1 to 100, CG yields the strongest and most monotonic improvements in PSNR/SSIM, outperforming other methods (Fig. 3). Increasing $K$ from 100 to 200 further improves PSNR/SSIM at $R$=10, while increasing runtime by only $\approx$0.1 s per image. In practice, we treat $K$ as a test-time budget (maximum number of CG updates) and enforce residual-based early stopping (tolerance of $10^{-8}$), so that reconstructions can terminate before the maximum.

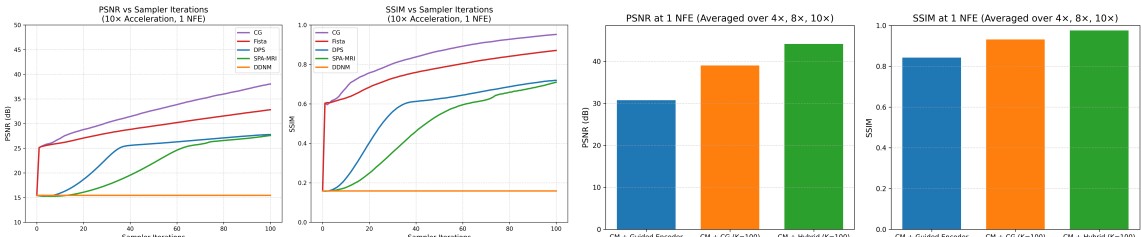

Figure 3: Ablation study across T1/T2. Left: Results of increasing $K$ for different DC schemes at $10\times$ (1 NFE, no guided encoding). CG (purple) shows largest first $K$ gain and remains best at $K=100$. Right: Effect of C-MORE components; hybrid (green): guided encoding + K-CG shows highest PSNR/SSIM.

We also specifically evaluate the full C-MORE framework with $K \in [25, 50, 100, 200, 300]$. Results show monotonic gains as $K$ increases (see Appendix B), but with diminishing returns and a near-plateau for low accelerations ($4\times$) around $K \approx 100$. Higher acceleration factors ($8\times$ and $10\times$) benefit more from larger $K$. Runtimes increase from $\approx 0.23$ s ($K=25$) to $\approx 0.52$ s ($K=300$) per image. Because $K$ is a maximum budget and early stopping is active, we report results for $K=100$ and $K=200$ as a practical configuration balancing sub-second runtime and reconstruction quality.

Finally, we compare three variants at NFE= 1: (a) CM with measurement-guided encoding only, (b) CM with CG at $K=100$ (no guided encoding), and (c) the full C-MORE hybrid (guided encoding + CG at $K=100$). Fig. 3 shows that the hybrid approach consistently achieves the highest average PSNR/SSIM across $R=4, 8, 10$, and improves with higher $K$. Training the encoder for longer (200,000 vs. 100,000 training iterations) also provides modest additional gains, and we use the longer-trained encoder for qualitative evaluation. This indicates that guided encoding and physics-based refinement play complementary roles.

**Cross-Anatomy Generalisation Evaluation:** To assess cross-domain generalisability, we evaluate the C-MORE model trained on the CMR×Recon cardiac T1/T2 dataset, with no finetuning or retraining, on an out-of-distribution (OOD) test set of 1,125 fastMRI knee slices. Reconstructions are performed using different masks across slices, which differ from the cardiac sampling patterns.

Table 2 reports PSNR, SSIM, NFE, and per-slice runtime across accelerations $R \in \{4, 8, 10\}$. C-MORE exhibits strong cross-anatomy generalisation despite never having seen knee data during training. At $R=4$, C-MORE ($K=200$) achieves PSNR 40.96 dB and SSIM 0.9527, outperforming all teachers and baselines, while remaining sub-second per slice (vs. $34.86-70.87$ s for the teacher and other methods). At higher accelerations, where the OOD shift becomes more pronounced and the reconstruction task is intrinsically more ill-posed, C-MORE's absolute fidelity naturally decreases—an expected outcome when transferring a cardiac-trained model to a substantially different anatomy under extreme undersampling. Crucially, however, C-MORE still surpasses all teachers and baselines by a clear margin at $8\times$ and $10\times$ accelerations, and it does so with only one NFE and orders-of-magnitude faster runtime. These results highlight that even under challenging OOD and high acceleration conditions, C-MORE remains the best performing and most efficient method, transferring to other anatomies beyond the cardiac domain.

Table 2: PSNR, SSIM, NFE, and runtime for fastMRI Knee Dataset at $4\times$, $8\times$, and $10\times$ undersampling. All models are evaluated out-of-distribution. Values are mean $\pm$ std. Best results are in **bold**; second-best are underlined.

| Acc | Method | NFE | PSNR ↑ | SSIM ↑ | Runtime (s) ↓ |
|---|---|---|---|---|---|
| $4\times$ | DPS | 500 | $24.70 \pm 5.17$ | $0.5923 \pm 0.1292$ | $70.87 \pm 42.76$ |
| | DDNM | 100 | $39.45 \pm 2.25$ | $0.9261 \pm 0.0269$ | $35.72 \pm 4.75$ |
| | DDS (Teacher, K=5) | 100 | $39.48 \pm 2.26$ | $0.9266 \pm 0.0268$ | $34.86 \pm 4.21$ |
| | Optimised Teacher (K=20) | 25 | $40.03 \pm 2.20$ | $0.9378 \pm 0.0216$ | $3.61 \pm 0.38$ |
| | C-MORE (K=100) | 1 | $39.98 \pm 2.00$ | $0.9404 \pm 0.0173$ | $\mathbf{0.76} \pm 0.17$ |
| | C-MORE (K=200) | 1 | $\mathbf{40.96} \pm 1.94$ | $\mathbf{0.9527} \pm 0.0140$ | $1.06 \pm 0.29$ |
| $8\times$ | DPS | 500 | $24.60 \pm 4.81$ | $0.5804 \pm 0.1225$ | $120.27 \pm 38.97$ |
| | DDNM | 100 | $34.67 \pm 2.80$ | $0.8425 \pm 0.0536$ | $46.67 \pm 3.03$ |
| | DDS (Teacher, K=5) | 100 | $34.63 \pm 2.80$ | $0.8422 \pm 0.0536$ | $45.32 \pm 2.23$ |
| | Optimised Teacher (K=20) | 25 | $35.49 \pm 2.59$ | $0.8607 \pm 0.0469$ | $4.03 \pm 0.38$ |
| | C-MORE (K=100) | 1 | $36.19 \pm 2.26$ | $0.8740 \pm 0.0369$ | $\mathbf{0.59} \pm 0.05$ |
| | C-MORE (K=200) | 1 | $\mathbf{36.83} \pm 2.24$ | $\mathbf{0.8887} \pm 0.0328$ | $0.80 \pm 0.25$ |
| $10\times$ | DPS | 500 | $24.62 \pm 4.61$ | $0.5811 \pm 0.1184$ | $123.56 \pm 39.79$ |
| | DDNM | 100 | $33.59 \pm 2.81$ | $0.8145 \pm 0.0610$ | $47.04 \pm 3.12$ |
| | DDS (Teacher, K=5) | 100 | $33.55 \pm 2.81$ | $0.8142 \pm 0.0610$ | $45.66 \pm 2.43$ |
| | Optimised Teacher (K=20) | 25 | $34.59 \pm 2.63$ | $0.8362 \pm 0.0546$ | $3.64 \pm 0.39$ |
| | C-MORE (K=100) | 1 | $35.57 \pm 2.31$ | $0.8546 \pm 0.0435$ | $\mathbf{0.58} \pm 0.05$ |
| | C-MORE (K=200) | 1 | $\mathbf{36.06} \pm 2.29$ | $\mathbf{0.8673} \pm 0.0395$ | $0.80 \pm 0.23$ |

Notably, this performance is achieved *without* any exposure to knee data, suggesting that modest finetuning on the target anatomy would close the remaining gap at higher accelerations and further amplify C-MORE's advantage. Example reconstructions are shown in Appendix C.

## 4. Conclusion

We have introduced the first one-step consistency model framework for accelerated MRI reconstruction, uniting the power of diffusion generative priors with the efficiency of a single-pass network. Our method achieves high-fidelity reconstructions in a single NFE while rigorously enforcing measurement constraints. Across diverse settings, we have demonstrated that C-MORE consistently outperforms state-of-the-art diffusion-based samplers, while achieving $\approx 22 - 193\times$ speed-up in just 1 NFE, compared to diffusion baselines requiring hundreds of NFEs. We have further demonstrated that C-MORE surpasses strong non-diffusion baselines, without requiring multiple models for each setting, and generalises to unseen data from another anatomy. This work therefore establishes a flexible, real-time reconstruction pipeline that potentially transforms a slow capability into a clinically plausible, on-the-fly solution. Future work will investigate downstream tasks and prospective validation to strengthen translational impact. Ultimately, C-MORE offers a promising blueprint for achieving state-of-the-art reconstruction quality with real-time performance.

## Acknowledgments

This work was supported in part by the UK Engineering and Physical Sciences Research Council (EPSRC) grants under TrustMRI [EP/X039277/1]. Computational resources for some experiments were also provided through the Fetch.ai Compute Grant Award by Fetch.ai Innovation Lab.

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

## Appendix A. Reconstructions across a full T1 and T2 parametric mapping sequence

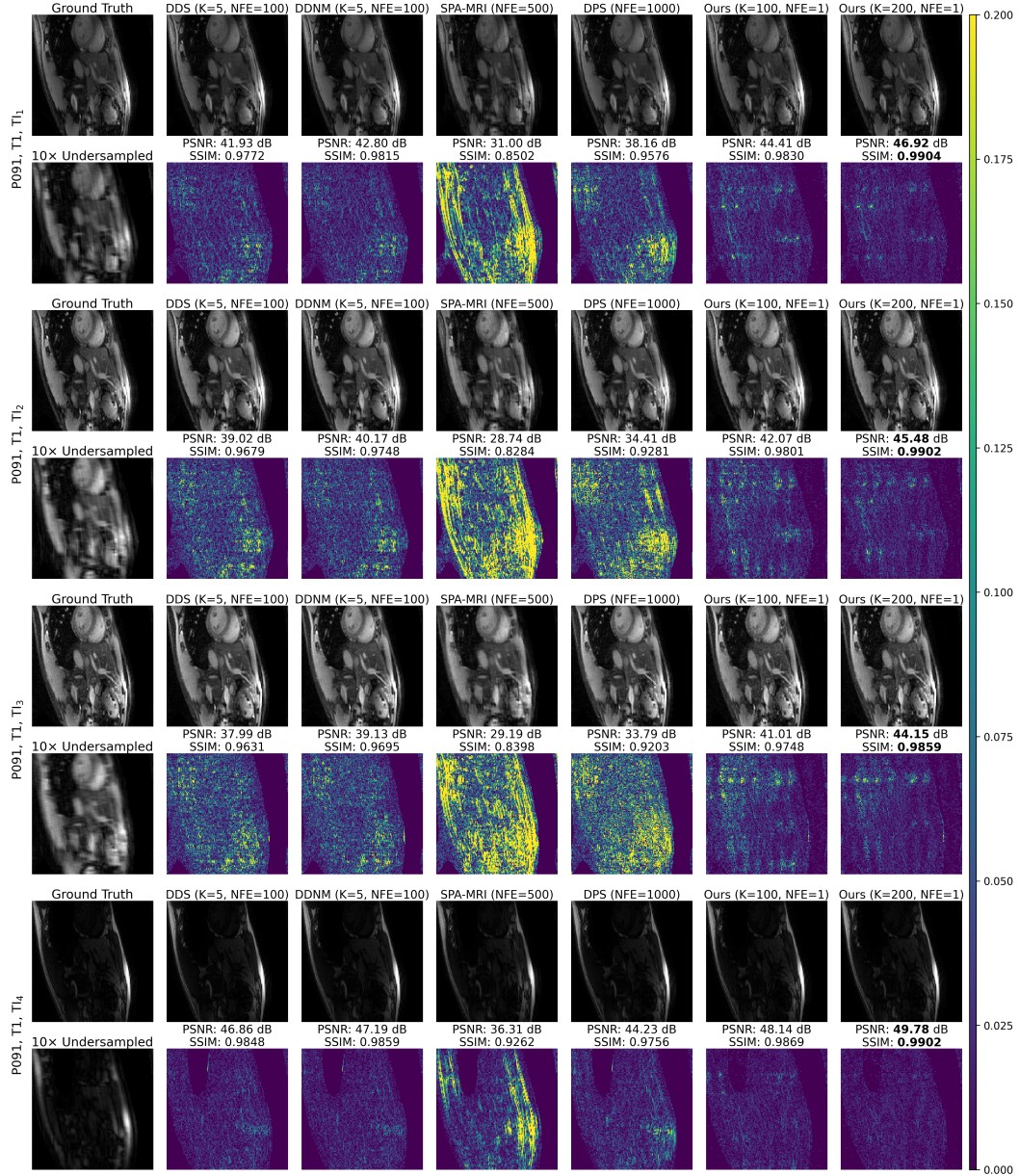

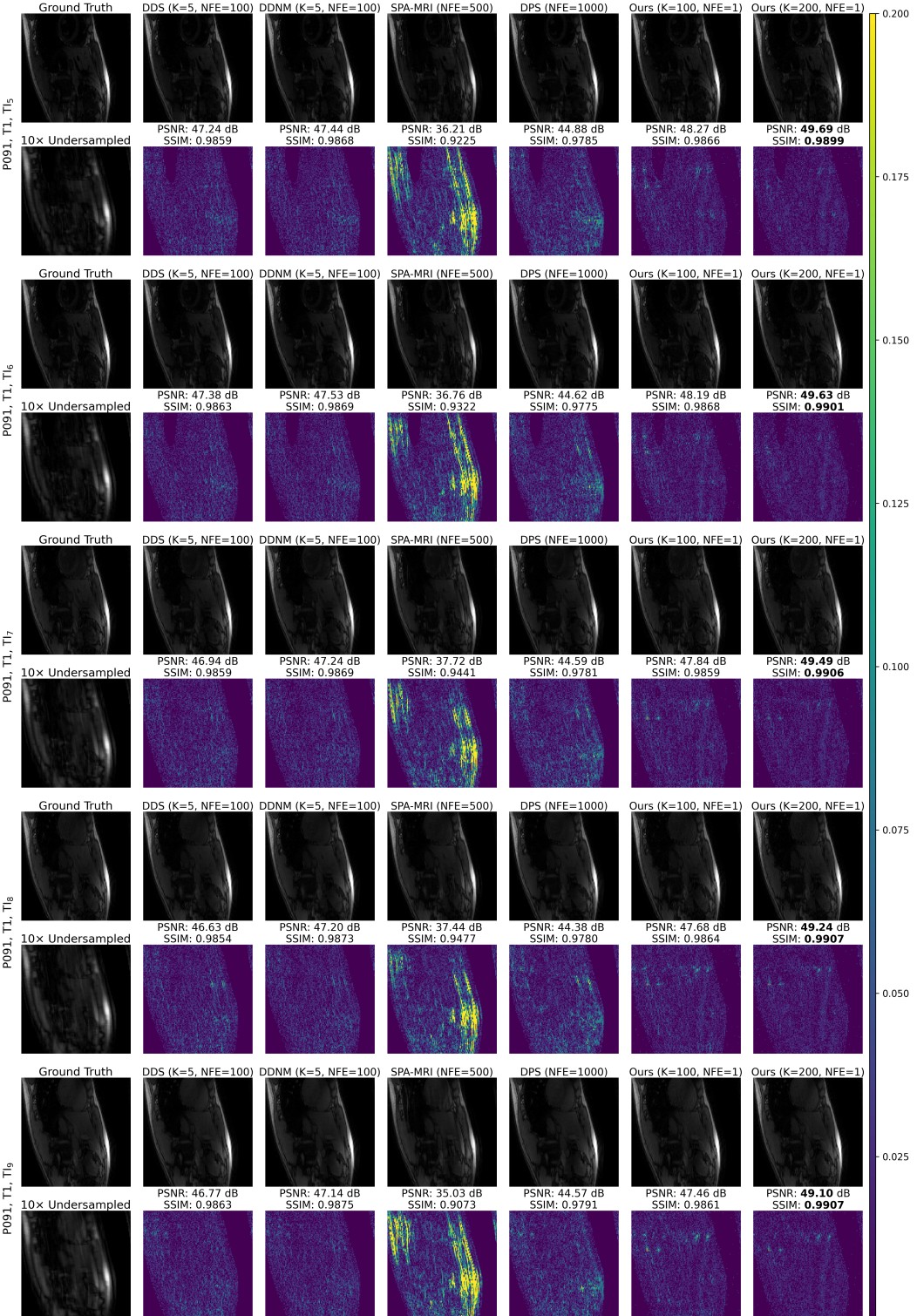

Figure 4: Reconstructions across the full T1 parametric mapping sequence (9 inversion times) for patient 91 at 10× acceleration.

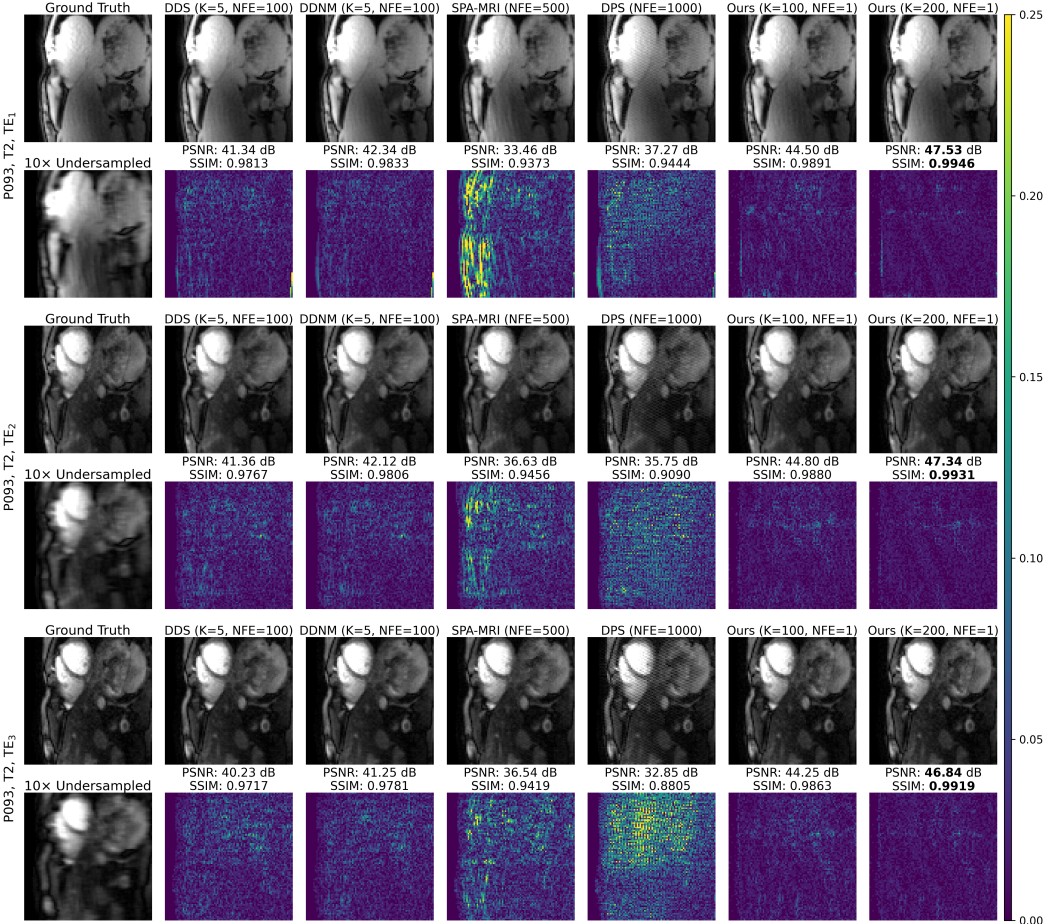

Figure 5: Reconstructions across the full T2 parametric mapping sequence (3 echo times) for patient 93 at 10× acceleration.

## Appendix B. PSNR/SSIM vs. CG Budget K

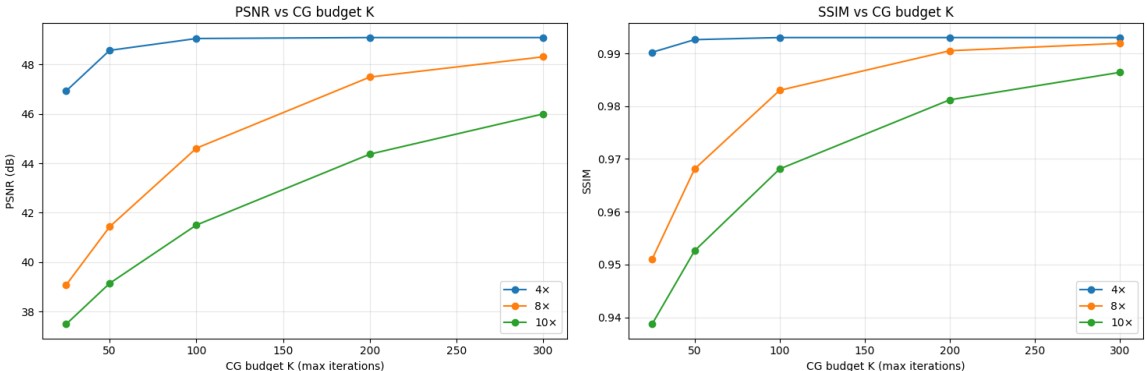

Figure 6: Results of increasing $K$ for C-MORE across acceleration factors. Performance improves monotonically with K, but shows diminishing returns; early stopping makes K a maximum budget rather than a fixed per-image iteration count.

## Appendix C. Cross-Anatomy Generalisation Reconstructions

We provide qualitative examples showing C-MORE reconstructions on an out-of-distribution fastMRI knee test set (1,125 slices). The single C-MORE model was trained only on CMR×Recon cardiac T1/T2 data and evaluated without retraining or finetuning. For each sample, we show: ground truth, zero-filled input, diffusion baselines (Teacher, DDNM, DPS, Optimised Teacher), and C-MORE (K=100 and K=200). The full quantitative breakdown for the knee experiments is reported in Table 2.

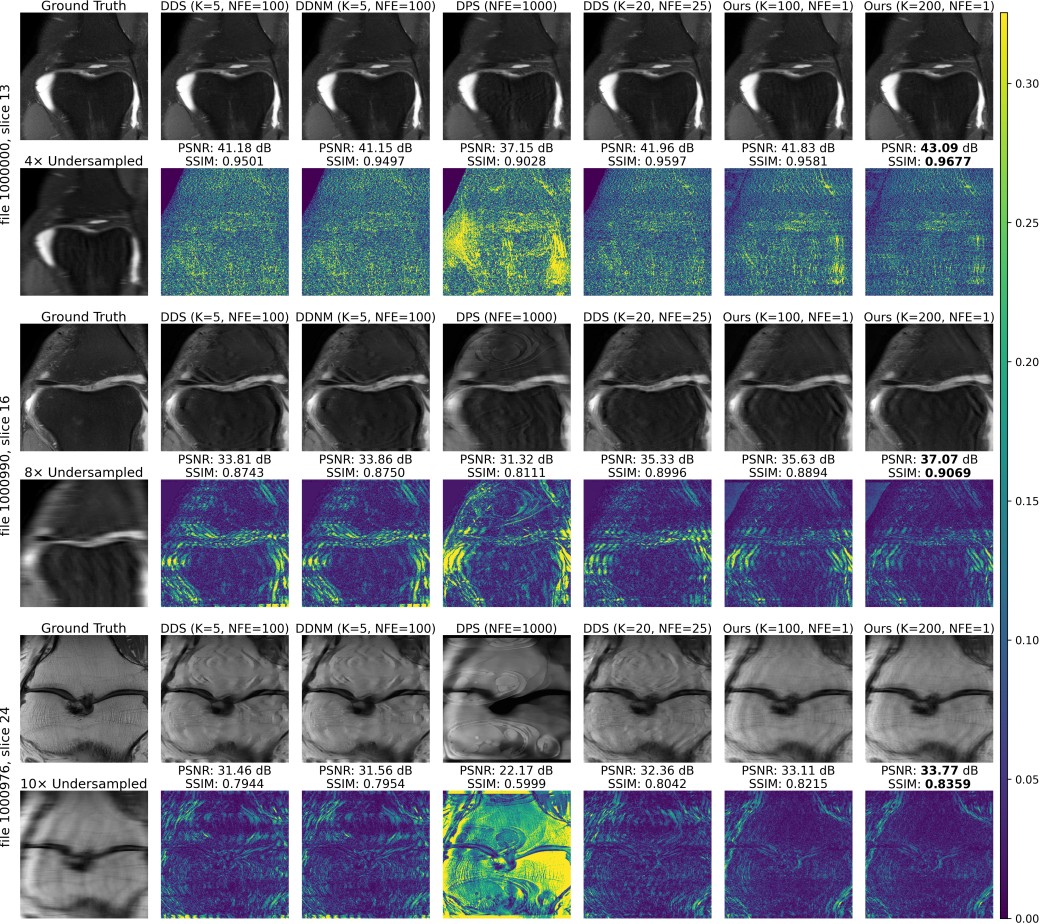

Figure 7: **Example out-of-distribution knee reconstructions.** Rows: three slices from the fastMRI knee test set. Each row shows the ground truth (left), zero-filled input (left on second row), error maps (second row), diffusion baselines (middle), and C-MORE outputs (right). All models are evaluated out-of-distribution. PSNR/SSIM are printed below each reconstruction. While fidelity generally degrades at 8× and 10× due to the challenging OOD setting and extreme undersampling, C-MORE still produces consistently higher-fidelity reconstructions than state-of-the-art diffusion baselines and does so with sub-second runtime. Further gains would be achievable with modest finetuning on the target anatomy.

