# OpenReview forum: "Beyond Diffusion: Consistency Models for One‑Step, High‑Fidelity MRI Reconstruction"
_MIDL.io/2026/Conference — MIDL 2026 Poster_

### Official Review · Reviewer_h83o · 2025-12-29

**Confidence:** 4
**Preliminary Rating:** 3
**Final Rating:** 3

**Summary:**

This paper proposes the use of a consistency model for one-step, high-fidelity MRI reconstruction, aiming to overcome the limitations of diffusion-based methods that require hundreds to thousands of neural function evaluations (NFEs) during inference. Experimental results demonstrate that the proposed approach can generate high-quality cardiac MRI images with significantly improved reconstruction efficiency.

**Strengths:**

The paper introduces the use of a consistency model for MRI reconstruction, substantially reducing volumetric image reconstruction time while maintaining high image quality.

The proposed method effectively addresses the efficiency bottleneck of diffusion-based reconstruction approaches.

**Weaknesses:**

The evaluation is conducted on only a single dataset, which limits the generalizability and reliability of the reported results.

Although the method targets improved practicality in clinical MRI reconstruction, no experiments are provided to assess the clinical utility of the reconstructed images.

**Detailed Comments:**

This paper explores the application of consistency models for MRI image reconstruction, with a clear advantage in reconstruction efficiency, as only a single-step NFE is required during inference. This represents a meaningful improvement over traditional diffusion-based approaches.

However, the experimental evaluation is limited to a single dataset, which weakens the evidence for the method’s robustness and general applicability. Furthermore, while the method is motivated by clinical deployment considerations, the study lacks experiments evaluating the clinical usability of the reconstructed images, such as performance in downstream diagnostic tasks or assessment by clinical experts.

**Justification Of Final Rating:**

I strongly recommend that the authors evaluate the generated data on at least one downstream task to better demonstrate its practical usability, rather than relying solely on image quality metrics. Since the primary purpose of generating medical images is to support research and downstream applications, assessing the usability of the generated data is an indispensable component of image generation evaluation.

**Justification Of The Preliminary Rating:**

The limited experimental evaluation on a single dataset and the absence of clinical utility assessment prevent a comprehensive understanding of the method’s robustness and real-world impact. Addressing these limitations would likely justify a higher rating.

**Questions To Address In The Rebuttal:**

If the authors can address the following points, the paper would be significantly strengthened:

Evaluate the proposed method on at least two representative and diverse datasets to demonstrate robustness and generalizability.

Include experiments that assess the clinical usability of the reconstructed images, beyond standard reconstruction quality metrics.

---

> ### Author Response · Authors · 2026-01-24
> **Reviewer h83o: Responses to Weaknesses (W), Detailed Comments (DC), and Questions (Q)**
>
> **Re: Evaluation on more than one dataset (W1, DC1, Q1)**
>
> Thank you for this important suggestion. To address this concern, we have conducted an out‑of‑distribution (OOD) evaluation to demonstrate the model’s generalisability to other datasets. Specifically, we applied the same C‑MORE model (trained only on cardiac T1/T2 data) to the **fastMRI knee dataset** (1,125 slices), without any retraining or fine‑tuning. This setting tested generalisation on a completely different anatomy (knee vs. cardiac) and was collected under a different institution. Across accelerations 4×, 8×, and 10×, **C‑MORE consistently outperformed both the teacher and all diffusion baselines on the unseen knee dataset**, achieving higher PSNR/SSIM while maintaining sub‑second runtime. We have added a dedicated subsection and table summarising these results.
>
> ---
>
> **Re: Clinical usability experiment and addressing real-world impact (W2, DC2, Q2)**
>
> 1. We appreciate this thoughtful point. Our work at its current stage focuses on a foundational challenge in accelerated MRI: **achieving state‑of‑the‑art generative reconstruction quality at real‑time speed**. This capability is a key enabling step toward clinical translation, as existing diffusion‑based methods remain too slow for routine use. Evaluating downstream utility is a natural next step; however, considering the scope and space constraints of the submission, we leave this to future work.
>
> 2. Furthermore, to better demonstrate C-MORE's potential real-world impact for accelerated MRI reconstruction, we have further expanded the paper to add comparisons to MoDL, a widely used non‑diffusion unrolled baseline that is considered a practical clinical competitor (results in Table 1). C‑MORE achieves **substantially higher fidelity** than MoDL across accelerations, while maintaining competitive runtime and requiring only a single model across contrasts, coil settings, and acceleration factors. This indicates that C‑MORE’s combination of high fidelity, generalisability, and sub‑second runtime makes it particularly promising for real‑time clinical workflows.
>
> 3. We have also revised the Conclusion to explicitly state that prospective evaluation is an important future direction.

---

> ### Author Response · Authors · 2026-01-29
> **Follow-up Response to Reviewer's Suggestion/Justification Of Final Rating**
>
> Thank you for your suggestion. Based on your feedback, we explored a downstream T1/T2 mapping evaluation; however, clinically meaningful mapping requires slice-level acquisition metadata and ROI/segmentation annotations that are not available in our current data split. Reporting partial or ad-hoc maps without these elements would risk misleading conclusions. So, to address your feedback as responsibly as we can, we commit to a downstream mapping evaluation as future work and have now stated this explicitly in the Conclusion.
>
> We also note that, in response to your earlier concern about generalisability, we added an out-of-distribution evaluation on the fastMRI knee dataset showing that the same C-MORE model (without finetuning or retraining) transfers across anatomy and outperforms the compared methods.
>
> We believe the study provides strong value by addressing a key practical bottleneck (achieving diffusion-level reconstruction quality at real-time speed), which is an important prerequisite for the downstream and clinical evaluations you highlight. We hope these additions and clarifications help address your concerns and better highlight the generalisability and practical significance of the proposed approach.

---

### Official Review · Reviewer_sqmk · 2026-01-09

**Confidence:** 3
**Preliminary Rating:** 4
**Final Rating:** 4

**Summary:**

This work deals with a consistency model based approach to accelerated MRI reconstruction. The authors demonstrate that their approach is able to provide superior image quality at significantly lower computational cost when compared to current state of the art diffusion based sampling approaches to deep learning based reconstruction

**Strengths:**

Its a well written paper and generally easy to follow. The presented results are quite compelling and deserve broader discussion within the community. I have some minor comments that I expect the authors to address readily

**Weaknesses:**

See comments below. See comments below. See comments below. See comments below. See comments below. See comments below. See comments below. See comments below. See comments below. See comments below. See comments below.

**Detailed Comments:**

The repeatedly changing error scales on the plots really makes it hard to compare across different setups. While I understand a single range wont work, please consider limiting it to a few different error scale ranges so that error maps are easier to evaluate

The DPS reconstruction seem to exhibit a strange looking cross hatch artifact in Fig 4. Please comment

For the sake of completeness, it would be good to demonstrate the impact of the different methods on the T1 and T2 maps themselves

Please add some more detail on the sub-sampling approach. Unclear if the same sub-sampling is being used across the different contrasts. Also unclear if the acquisition is decoupled across the TE/TI's or if  it would support joint reconstruction. Could be worth adding a few comments related to it

**Justification Of Final Rating:**

The authors have put substantial effort in the revision which is greatly appreciated. They have also updated the manuscript suitably to address all of my feedback - I am happy to recommend acceptance.

**Justification Of The Preliminary Rating:**

Paper is well written and explores novel ideas that can enhance cardiac imaging workflow. I expect the authors to be able to update the draft readily to account for my feedback after which I can recommend acceptance

**Questions To Address In The Rebuttal:**

Most of my comments can likely be addressed with minor updates to figures / discussion sections.

---

> ### Author Response · Authors · 2026-01-24
> **Reviewer sqmk: Responses to Detailed Comments (DC)**
>
> **Re: Error scale ranges in visualisations (DC1)**
> > “...please consider limiting it to a few different error scale ranges so that error maps are easier to evaluate.”
>
> Thank you for pointing that out. We have adjusted our figures to use a consistent scale per figure (see Figures 2, 3, and 4) to improve visual comparability across setups.
>
> ---
>
> **Re: DPS cross-hatch artefacts in Figure 4 (DC2)**
> > “The DPS reconstruction seems to exhibit a strange looking cross hatch artifact in Fig 4. Please comment.”
>
> Thank you for highlighting this. The cross-hatch artefacts in DPS arise from its repeated gradient steps of the data-consistency term during sampling. Under Cartesian undersampling, this operator induces structured aliasing. When repeatedly applied over long trajectories (e.g., high iteration counts), these alias-correlated residuals accumulate into lattice-like artefacts. We verified that reducing DPS iterations (e.g., from 1000 to 500 NFEs) substantially reduces this observed artefacts.
>
> For quantitative evaluation and runtime comparison, we use DPS at 500 NFEs (see Table 1) and now also use the same in Figure 2 for consistency. However, for completeness, we also show DPS at 1000 NFEs in the appendix (Figures 3 and 4) to reinforce our method’s superiority against DPS even with higher iteration counts.
>
> ---
>
> **Re: Impact on T1/T2 maps (DC3)**
> > “For the sake of completeness, it would be good to demonstrate the impact of the different methods on the T1 and T2 maps themselves.”
>
> Thank you for the suggestion. Our work at its current stage focuses on a foundational challenge in accelerated MRI: **achieving state‑of‑the‑art generative reconstruction quality at real‑time speed**. This capability is a key enabling step toward clinical translation, as existing diffusion‑based methods remain too slow for routine use. Evaluating downstream utility is a natural next step; however, considering the scope and space constraints of the submission, we leave this to future work.
>
> ---
>
> **Re: Subsampling details and joint reconstruction (DC4)**
> > “Please add some more detail on the sub-sampling approach... Also unclear if the acquisition is decoupled across the TE/TI’s or if it would support joint reconstruction. Could be worth adding a few comments related to it.”
>
> Thank you for your valuable feedback. We have added a clarification to the Dataset subsection. Specifically:
> 1. **Subsampling approach**: We considered acceleration factors 4x, 8x, and 10x with uniform Cartesian undersampling masks provided with the dataset. These masks are identical across contrasts for a given acceleration factor.
> 2. **Regarding joint reconstruction**: all reconstructions in this work are performed per-contrast. However, the C-MORE architecture is compatible with joint reconstruction, and this is a straightforward extension we leave to future work.

---

> ### Comment · Area_Chair_5rHd · 2026-02-01
>
> Please provide your final rating
>
> AC

---

### Official Review · Reviewer_NgGo · 2026-01-10

**Confidence:** 4
**Preliminary Rating:** 4
**Final Rating:** 5

**Summary:**

The authors propose C-MORE (Consistency-Model-based One-step Reconstruction for MRI), claiming to be the first one-step consistency-model-based framework for accelerated MRI reconstruction. The method trains an unconditional EDM diffusion teacher, distills it into a one-step consistency model (CM) prior, utilizes measurement-guided encoding to condition on zero-filled reconstructions, and applies K conjugate gradient (CG) refinement updates to ensure data consistency. They worked on the CMRxRecon cardiac dataset across multiple contrasts and acceleration factors (4×, 8×, 10×). C-MORE claims to achieve higher PSNR/SSIM than diffusion-based baselines while running in 0.18-0.52 seconds (22-193× speedup) with just 1 NFE.

**Strengths:**

1. Adapting consistency models specifically for MRI reconstruction is interesting and relatively unexplored. The combination of consistency distillation with measurement-guided encoding and physics-based refinement is a reasonable design choice.

2. Table 1 shows C-MORE consistently achieving the best PSNR/SSIM across all T1/T2 contrasts and acceleration factors (4×, 8×, 10×), with substantial improvements over baseline methods.

3. Reducing reconstruction time from 11-35 seconds to 0.18-0.52 seconds is significant for clinical workflows. The sub-second runtime makes real-time reconstruction a plausible option.

4. A single C-MORE model handles both T1 (9 contrasts) and T2 (3 contrasts) mapping sequences without retraining, demonstrating good generalization across different cardiac contrasts.

5. Evaluation on CMRxRecon (a recent and real public dataset) with both single- and multi-coil acquisitions provides a reasonable testbed for cardiac MRI reconstruction.

**Weaknesses:**

1. While the authors claim in the paper about "one-step" reconstruction, it utilizes K=200 (or 100) iterations of CG refinement. It's true that CG is "physics-based" and not a "neural evaluation", but it still incurs significant computational overhead in terms of FFT operations. Calling this "one-step" is technically true for the model, but potentially misleading for the pipeline.

2. Methodology is a little hard to follow, especially the overview of the proposed C-MORE framework. The authors should come up with a more comprehensive figure with sufficient annotations.

3. The performance is strictly bound by the EDM teacher. If the teacher has inherent biases or hallucinations in high-acceleration regimes (e.g., 10x), the distilled student will likely inherit them.

4. The baseline comparison is portrayed as a story of "diffusion being slow". The authors compare against strong non-diffusion learned reconstructions (e.g., variational networks/unrolled methods) that already run in very little time, and are the real competition in clinics.

5. The authors used retrospective masking to create undersampled data from CMRxRecon dataset. While it might be a valid approach, the retrospective undersampling limits clinical validity. Prospective validation would strengthen the work.

**Detailed Comments:**

1. Comparing against a fast, strong unrolled baseline (VarNet-like) at matched runtime would be great.

2. Table 1 shows the Teacher (K=5 per NFE, 100 NFE = 500 total DC) achieving PSNR 48.44 at 4×, while C-MORE (K=200, 1 NFE = 200 total DC) achieves PSNR 49.06. This is certainly better, but is it because of the one-step CM design or because the DC updates are concentrated differently?

3. Computational cost breakdown would be interesting to check. Like what percentage of C-MORE's runtime is CM evaluation vs. K-CG updates vs. other operations?

4. Section 2.3 mentions using LPIPS for perceptual consistency distillation "to better preserve local structure" but provides no ablation comparing L2 vs. LPIPS distillation losses. How much does this design choice matter?

5. How does performance scale with different K values (e.g., K=25, 50, 300) across different acceleration factors? Is there a sweet spot?

**Justification Of Final Rating:**

I think my concerns have been addressed/ commented on. The author(s) carefully expanded the manuscript to compare against new baselines, and they also worked on their methodology figure.
So I am increasing the rating here. Thanks!

**Justification Of The Preliminary Rating:**

The paper presents solid empirical work in adapting consistency models to MRI reconstruction, yielding strong quantitative results and clinically significant speed-ups. However, it has a critical framing issue: repeatedly emphasizing "one NFE" and "eliminating multistep diffusion sampling" obscures that K=100-200 CG iterations are still data consistency steps, just concentrated after one network evaluation rather than distributed across many.
The comparison is slightly unfair because baselines use only K=5 DC per NFE, and a fair comparison matching the total DC budget (e.g., Teacher with 25 NFE × K=20) is missing. Evaluation is limited to retrospective undersampling on cardiac MRI, raising questions about generalization to prospective acquisition and other settings.
With major revisions addressing the presentation of the methods section, framing the narrative, making fairer comparisons, and providing a broader evaluation, this could be considered a good work.

**Questions To Address In The Rebuttal:**

Please refer to the Weakness and Detailed Comments section and respond/ comment on those issues.

---

> ### Author Response · Authors · 2026-01-24
> **Reviewer NgGo: Responses to Weaknesses (W) and Detailed Comments (DC)**
>
> **Re: “One-step” wording, CG cost and runtime breakdown (W1, DC3)**
> > “While the authors claim in the paper about ‘one-step’ reconstruction, it utilizes K=200 (or 100)...”
> > “Computational cost breakdown would be interesting to check...”
> > “However, it has a critical framing issue: repeatedly emphasizing ‘one NFE’ and ‘eliminating multistep diffusion...’”
>
> Thank you for your feedback.
>
> 1. **Clarifying “one-step” terminology**
>    By “one-step,” we refer to one NFE of the CM, consistent with terminology in diffusion/consistency literature. While CG involves multiple iterations, its computational overhead is marginal (see below). To balance clarity and consistency with prior work, we have revised the paper to clarify: “eliminating the need for *multi-NFE* diffusion sampling”—avoiding misinterpretation that this includes DC updates.
>
> 2. **Runtime breakdown**
>    CG stage takes ≈0.09–0.24 s for K=100 and ≈0.16–0.39 s for K=200. Total per-image runtime remains sub-second (0.18–0.52 s). We’ve now added a comment on this in the paper.
>
> ---
>
> **Re: Annotating the methodology overview figure (W2)**
>
> Thank you for your feedback. We’ve updated the overview diagram with additional annotations and the caption also clarifies the framework.
>
> ---
>
> **Re: Is performance bounded by teacher? (W3)**
>
> Thank you for your feedback. Our results show that C-MORE exceeds the teacher’s performance. For example, at R=4 (T1), the teacher (100 NFE, K=5) achieves PSNR 48.44 dB, while C-MORE (K=200) achieves 49.06 dB. Similar gains hold at R=8 and R=10 (see Table 1). We also evaluated an optimised teacher (25 NFE, K=20) and found C-MORE still achieves higher PSNR across all settings and higher SSIM in all but two cases, while maintaining single NFE and sub-second runtime.
>
> This performance is attributed to our use of measurement-guided encoding and physics refinement. Without these, we acknowledge that the distilled student could inherit teacher biases. This is why we trained a high-fidelity teacher (~750k epochs vs. 200k for student) and we note that future work could explore consistency training (vs. distillation) to further overcome this ceiling.
>
> ---
>
> **Re: Comparison to fast unrolled baselines used in clinics (W4, DC1)**
>
> Thank you for your valuable feedback. We expanded the manuscript to compare against MoDL, a representative fast unrolled model.
>
> 1. **MoDL results**
>    We trained separate MoDL models for 4× and 8× multi-coil settings and added results to Table 1. MoDL is fast (≈0.09 s/image) but requires separate models per setting and degrades substantially at 8× (e.g., T1: 35.91 dB vs. C-MORE: 47.20 dB). In contrast, a single C-MORE model handles multiple contrasts, accelerations, and coil settings without retraining, maintaining higher fidelity and subsecond runtime between 0.18–0.52 s/image.
>
> 2. **Narrative clarification**
>    Our goal is not just speed, but also quality and generalisability. Diffusion models advance reconstruction fidelity and robustness beyond unrolled baselines, especially at high accelerations. C-MORE retains these advantages while delivering real-time speed.
>
> ---
>
> **Re: Retrospective vs. prospective undersampling (W5)**
>
> Thank you for your feedback. The current dataset that can be accessed are retrospective undersampled data, which have been widely used in various SOTA and benchmark studies. We follow this standard practice to validate our approach and benchmark our results. We agree that prospective evaluation is important and have revised the Conclusion to state this as future work.
>
> ---
>
> **Re: CM design vs. DC concentration and Teacher (25 NFE × K=20) (DC2)**
> > “Is it because of the one-step CM design or DC updates?”
>
> Thank you for your feedback. We ran the requested Teacher (25 NFE × K=20) and added the results to Table 1. C-MORE still achieves higher PSNR and SSIM (except T1 SSIM at 4× and 10×, where it’s a close second), while maintaining single NFE and subsecond runtime. This Optimised Teacher is slower (≈2.3 s/image). Thus, C-MORE’s improvement is not solely due to DC concentration, but also the one-step guided CM design.
>
> ---
>
> **Re: LPIPS vs. L2 (DC4)**
>
> We chose LPIPS because (i) prior CM work shows LPIPS outperforms L1/L2 by a large margin over training iterations (Song et al., 2023, Fig. 3), and (ii) we observed the same trend during training. We’ve added a comment and citation to justify this choice.
>
> ---
>
> **Re: Sweet spot for K (DC5)**
>
> Thank you for your question. We treat K as a maximum CG budget and use residual-based early stopping. Experiments with K ∈ [25,50,100,200,300] show monotonic PSNR improvement (41.17 → 47.80 dB), with diminishing returns beyond K≈100 for low accelerations. Higher accelerations benefit more. Runtime increases from ≈0.23 s (K=25) to ≈0.52 s (K=300). We report K=100 and K=200 as practical operating points and have added comments to ablation and plots to the appendix showing PSNR/SSIM vs. K.

---

> ### Comment · Area_Chair_5rHd · 2026-02-01
>
> Please provide your final rating
>
> AC

---

### Author Rebuttal · Authors · 2026-01-25

**Rebuttal:**

We sincerely thank all reviewers for their thoughtful and constructive feedback. We have carefully addressed every concern and substantially strengthened the paper in response. Below we summarise the key revisions.

**1. New Generalisability Evaluation on a Second, Distinct Dataset**
To address generalisability concerns, we conducted a new Out‑of‑Distribution (OOD) test on the unseen fastMRI knee dataset, using the same C‑MORE model trained only on cardiac T1/T2 data.
- This test spans a completely different anatomy and was collected under a different institution.
- **C‑MORE outperforms the teacher and all baselines** across accelerations, achieving higher PSNR/SSIM (e.g., SSIM 0.9527 at 4x) and subsecond runtime. A subsection, table and appendix have been added.

**2. Added Comparison to Non-diffusion Unrolled Method**
Reviewers highlighted the need to position C-MORE relative to methods currently for clinical use.
- We added MoDL, a widely used unrolled method, and trained separate 4× and 8× models.
- While MoDL is faster (≈0.09s), C‑MORE achieves **substantially higher fidelity** across accelerations (e.g., 47.20 dB vs. 35.91 dB MoDL for T1 8×), while maintaining subsecond runtime (0.18-0.52s) and requiring only one model across contrasts, coil settings, and accelerations.
- This clarifies C‑MORE’s potential real‑world impact.

**3. Additional Experiments and Clarifications**
We added requested analyses, including:
- Teacher with 25 NFE × K=20 (C‑MORE still outperforms, while being >4× faster).
- Ablation on K with performance/runtime curves.
- Consistent error‑scale visualisations.
- Justification for LPIPS based on prior CM work and our own observations.

---

Overall, we appreciate the reviewers’ recognition of the novelty and impact of the work. With the new generalisability evaluation, expanded clinical‑relevant baseline comparisons, clarified methodology annotation, and additional analyses, we believe the revised paper now provides stronger evidence of C‑MORE’s generalisability and practical significance. **We further address each reviewer's comments point by point in more details below**. We hope these substantial improvements address all concerns and support a positive recommendation.

**Supporting Material:**

/attachment/e943cc8f685f7c4a7a689ad99a41cbbb29597bc6.pdf

---

### Meta-Review · Area_Chair_5rHd · 2026-02-03

**Recommendation:** Accept (Oral)
**Confidence:** 5

**Metareview:**

All reviewers agree on the importance and efficacy of the method. Major concerns, such as generalization performance, were addressed in the rebuttal. The remaining concern is the lack of utility experiments, which none of the reviewers consider significant enough to warrant a low score. Therefore, I recommend acceptance.

---

### Decision · Program_Chairs · 2026-02-13

Accept (Poster)